# Significance of NatB-mediated N-terminal acetylation of auxin biosynthetic enzymes in maintaining auxin homeostasis in *Arabidopsis thaliana*

Hai-Qing Liu[1,5], Zuo-Xian Pu[1,2,5], Dong-Wei Di[1,3], Ya-Jie Zou[1,2], Yu-Man Guo[1,2], Jun-Li Wang[1,2], Li Zhang[1,4], Peng Tian[1,2], Qiong-Hui Fei[1,2], Xiao-Feng Li[1,2], Allah Jurio Khaskheli[1,2], Lei Wu ®[1,2 ✉] & Guang-Qin Guo ®[1,2 ✉]

The auxin IAA (Indole-3-acetic acid) plays key roles in regulating plant growth and development, which depends on an intricate homeostasis that is determined by the balance between its biosynthesis, metabolism and transport. YUC flavin monooxygenases catalyze the rate-limiting step of auxin biosynthesis via IPyA (indole pyruvic acid) and are critical targets in regulating auxin homeostasis. Despite of numerous reports on the transcriptional regulation of *YUC* genes, little is known about those at the post-translational protein level. Here, we show that loss of function of CKRC3/TCU2, the auxiliary subunit (Naa25) of *Arabidopsis* NatB, and/or of its catalytic subunit (Naa20), NBC, led to auxin-deficiency in plants. Experimental evidences show that CKRC3/TCU2 can interact with NBC to form a NatB complex, catalyzing the N-terminal acetylation (NTA) of YUC proteins for their intracellular stability to maintain normal auxin homeostasis in plants. Hence, our findings provide significantly new insight into the link between protein NTA and auxin biosynthesis in plants.

[1] MOE Key Laboratory of Cell Activities and Stress Adaptations, School of Life Sciences, Lanzhou University, Lanzhou 730000, China. [2] Gansu Province Key Laboratory of Gene Editing for Breeding, School of Life Sciences, Lanzhou University, Lanzhou 730000, China. [3] Institute of Soil Science, Chinese Academy of Sciences, Nanjing 210008, China. [4] Basic Forestry and Proteomics Research Center, College of Life Sciences, Fujian Agriculture and Forestry University, Fuzhou 350002, China. [5] These authors contributed equally: Hai-Qing Liu, Zuo-Xian Pu. ✉email: leiwu@lzu.edu.cn; gqguo@lzu.edu.cn

As an important plant hormone, auxin plays a central role in the growth and development of plants[1–4]. Based on biochemical and genetic evidences, the major natural auxin, IAA (Indole-3-acetic acid), is mainly synthesized from tryptophan (Trp) via IPyA (Indole pyruvic acid) pathway[5–7] through two successive biochemical reactions. In this pathway, the TAA (TRYPTOPHAN AMINOTRANSFERASE OF ARABIDOPSIS) family proteins catalyze the conversion of Trp to IPyA[8–11], followed by YUC (YUCCA) flavin monooxygenase-like protein catalyzed conversion of IPyA to IAA[5,6,12,13].

During plant growth and development and response to environmental stimuli, auxin biosynthesis is regulated at multiple levels, including epigenetic regulation[14], transcriptional initiation[12,15], and post-translational protein modifications[16]. Among diverse post-translational modifications, N-terminal acetylation (NTA) is a highly abundant protein modification in eukaryotes in which the acetyl group of acetyl-CoA is transferred to the amino group of a protein at its N-terminus under the catalysis of N-terminal acetyltransferases (NATs)[17,18], altering the steric or chemical properties of the modified N-terminus that may affect protein–protein interaction[19,20], subcellular localization[21,22], protein folding and aggregation[23,24], stability or degradation[25–27] in normal cellular life in growth, development, and responses.

Although universally prevalent in eukaryotes, the functional significance of NTA has only recently emerged in plants, with a limited number of reports on its roles in growth, flowering, reproduction[28,29], stress[30] and immunity[27]. However, functional significance of NTA in auxin biosynthesis has not been reported so far. Here, we show evidences that the Arabidopsis NatB catalyzed NTA of YUC proteins can stabilize their intracellular stability to maintain normal auxin homeostasis in plants. Hence, our findings provide significant new insight into the link between protein NTA and auxin biosynthesis in plants.

## Results and discussion

**ckrc3 is an auxin-deficient mutant.** Auxin homeostasis in plants is crucial to control many aspects of plant growth and development[1,31]. To study its regulation, novel auxin-deficient mutants are vital for uncovering the unknown genes functioning in auxin homeostasis. However, the complexity of the auxin biosynthesis and metabolic pathways has greatly hindered the efforts to dissect their molecular mechanisms. To overcome this problem, a large-scale screening for auxin-deficient mutants has previously performed in our lab, in which the ckrc3 mutant was isolated as one of the so-called group II cytokinin (CK) induced root curling (ckrc) mutants[32]. When grown on medium containing 0.1 μM trans-zeatin (tZ), these mutants displayed a root curling phenotype, which is usually caused by auxin deficiency[11]. Besides some general pleiotropic developmental defects, such as the significantly reduced growth rate (Fig.1f–g; Supplementary Fig. 1d, i), leaf reticulation, early flowering, aborted ovules in short silique (Supplementary Fig. 1b, c, f, h[33]), impaired apical hooks (Supplementary Fig. 1e, g), ckrc3 mutant also exhibited typical auxin-deficient phenotypes in roots, including the reduced length of primary roots, wavy or CK-induced curling growth, defective gravitropic response (Fig. 1a–e), less number of root hairs (Supplementary Fig. 1a), weaker expression of the auxin reporters DR5::GUS or DR5::GFP (Fig. 1h, i) and reduced endogenous contents of free IAA and its metabolites (IAGlu, IAAsp, and especially oxIAA)[32], which can be rescued either by exogenous application or endogenous production (by superroot2, sur2 mutation) of IAA (Fig. 1a, c, d, j; Supplementary Fig. 2). Previously it was known that the defects in glucosinolate biosynthesis by the sur2 mutation can lead to auxin accumulation[34,35], which can be used to rescue auxin deficiency[8].

**CKRC3 gene encodes the auxiliary subunit which interacts with NBC to form a NatB complex for maintaining endogenous auxin levels.** Genetic analyses and map-based cloning identified ckrc3 as a loss of function mutation in the AT5G58450 gene. The mutation caused a premature termination by a G - > A transition, changing the tryptophan (at the 731th aa. position of the encoded protein) codon TGG into the TAG stop codon (Supplementary Fig. 3a–d). Genetic allelic analyses and molecular complementation confirmed that AT5G58450 is the CKRC3 gene (Supplementary Fig. 3e–g).

CKRC3 is annotated to encode the auxiliary subunit (Naa25) of a putative NatB in Arabidopsis, and was named TCU2 previously[33]. NatB is one of the eight NATs, NatA through NatH found so far in eukaryotes[36,37], specifically catalyzing the acetylation of proteins beginning with iM-D/E/N/Q[38–40].

The Arabidopsis NatB is presumed to contains a catalytic subunit Naa20 (NatB catalytic subunit, NBC) and an auxiliary subunit Naa25 (CKRC3/TCU2)[33]. In Candida albicans, Naa25 forms a horse-shoe-like deck to hold specifically its catalytic subunit Naa20[40]. The Arabidopsis Naa25/CKRC3/TCU2 protein has 1065 amino acids and contains several tetratricopeptide repetitions (TPRs) in its N-terminal part. TPR domains are composed of 3-16 degenerate tandem repeats of 34 amino acids that form structural domains in proteins that facilitate the assembly of large protein complexes[41,42]. CKRC3/TCU2 also contains a NatB domain with unknown function in the intermediate region, which is very conserved and occupies half of the NatB auxiliary subunit in all organisms studied (Supplementary Fig. 4a). As predicted, a strong interaction between CKRC3/TCU2 and NBC was detected by both yeast two-hybrid (Y2H) (Fig. 2a) and Co-IP assays in tobacco cells (Fig. 2b; Supplementary Fig. 14), which depends on both the TPR motif and NatB domain, as shown by experiments with different truncations of CKRC3/TCU2 (Supplementary Fig. 4), illustrating that CKRC3 and NBC can interact to form NatB complexes both exo and in planta, as also reported elsewhere[43] after the initial submission of this work. These results are consistent with the cytoplasmic co-localization of CKRC3/TCU2 and NBC, as revealed by CKRC3/TCU2-eGFP and NBC-GFP fusion proteins (Supplementary Fig. 5), and the similar auxin-deficient phenotypes between ckrc3, nbc–1 and ckrc3 nbc–1 (Fig. 2c–j; Supplementary Fig. 2; Supplementary Fig. 6). Hence, the interaction between NAA25/CKRC3/TCU2 and NAA20/NBC to form a NatB complex is required for maintaining normal IAA levels in Arabidopsis.

**NatB complex catalyzes N-terminal acetylation of YUC8 for its intracellular stability to maintain auxin homeostasis.** Among the known auxin biosynthetic pathways, YUCs catalyze the rate-limiting step of auxin biosynthesis via IPyA and are critical targets in regulating auxin homeostasis[44,45]. In Arabidopsis, 8 of the 11 YUCs[46] possess NatB substrate signatures (ME…, Supplementary Fig. 7). Among YUCs, which are highly functional redundant[46], only the loss of function mutant of YUCCA8 (yuc8) had ckrc phenotype (Supplementary Fig. 8). Because of this phenotypic advantage, we focused YUCCA8 (YUC8) in further investigations.

To confirm that YUC8 is a target protein of NatB, we first examined the enzyme activity of NBC in vitro. We used synthesized oligo-polypeptides as the substrates. NTA modification was detected by LC-MS/MS and found to occur on both the positive control N-terminal peptide of SNC1 protein[27] and that of our YUC8(iME–) target, but not in the negative control of the mutated form of YUC8(iMAE–), in which an insertion of A between M and E could theoretically disrupt the NatB substrate

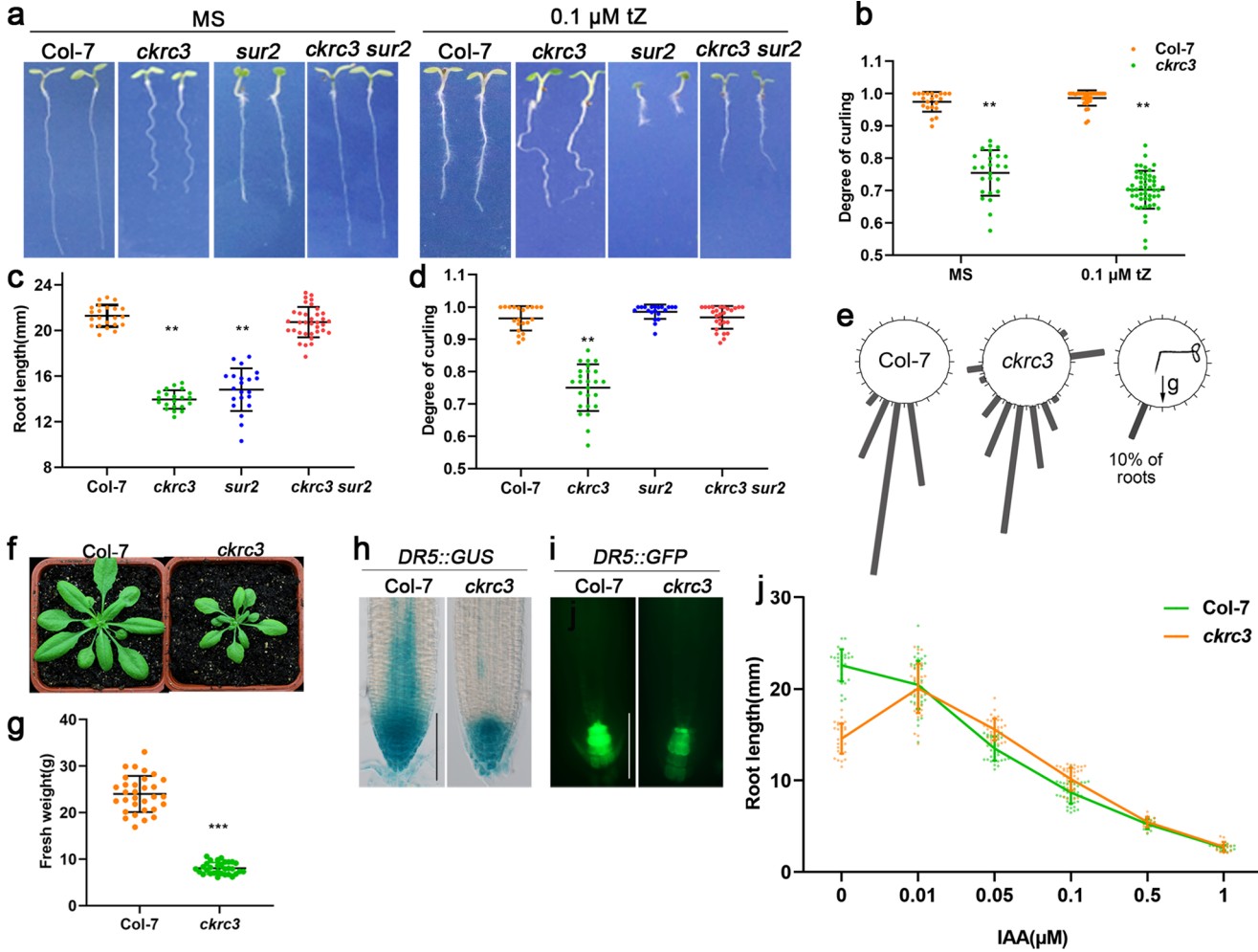

**Fig. 1 *ckrc3* is an auxin-deficient mutant. a–e** Plant phenotypes at 7 DAG (Day after germination) on MS media with or without 0.1 μM tZ (bar = 5 mm), quantification results of root length and degree of curling are shown in **b–d** (seedlings grow on MS in **c**, **d**) (*n* ≥ 20), bar = 5 mm. Data are presented as mean ± SD, \*\**P* < 0.01 according to ANOVA followed by Tukey's multiple comparison tests. **e** Quantification of root gravitropic responses (*n* ≥ 50). **f**, **g** Rosettes and fresh weight of 30 DAG plant (*n* = 31). Data are presented as mean ± SD, \*\*\**P* < 0.001 according to ANOVA followed by Tukey's multiple comparison tests. **h**, **i** GUS staining or GFP fluorescence of *DR5::GUS/GFP* marker lines, bar = 100 μm. **j** Primary root length at 7 DAG on MS medium with increasing concentrations of IAA (*n* ≥ 15). Data are presented as mean ± SD.

signature (Supplementary Fig. 9; Supplementary Tab. 1). The in vitro NTA of the potential N-terminal peptide substrates of other YUCs by NatB were also confirmed (Supplementary Tab. 1).

To detect the in vivo NatB-mediated NTA, we expressed YUC8$^{N7}$-YFP fusion protein in transgenic plants. $^{Ac}$iMet were detected in YUC8$^{N7}$-YFP by LC-MS/MS in the *yuc8* single mutant but not in *yuc8 ckrc3* double mutant background (Supplementary Fig. 12), confirming that the NTA of YUC8 were mediated by NatB complex.

To investigate the role of NatB-mediated NTA of YUC8 in auxin homeostasis in plants, we generated *CaMV 35S::YUC8-mGFP* transgenic lines in both wild-type (WT) (#1 and #13) and *ckrc3 nbc-1* (#3 and #7) backgrounds, and over-expression of *YUC8-mGFP* was obtained in all of these 4 independent transgenic lines (Fig. 3c), which produced high auxin phenotypes (long hypocotyl and epinastic cotyledons) in WT (Fig. 3a, b), indicating that the fusion protein is fully functional; but not in the *ckrc3 nbc-1* double mutant (Fig. 3a, b), revealing that NatB-mediated NTA of YUC8 is crucial for its function in auxin biosynthesis, most likely by promoting the stability of YUC8 protein, as reported elsewhere for SNC1 protein[27], which was

confirmed in our present study by transient expression of YUC8-eGFP fusion protein in *Arabidopsis* mesophyll protoplasts (Supplementary Fig. 10) and WB analysis on protein abundance in transgenic plants (Fig. 3d; Supplementary Fig. 15), where the transcript levels was comparable between WT and the *ckrc3 nbc-1* double mutant (Supplementary Fig. 13).

The substrate specificities of NATs are generally determined by the first two amino acids of the target protein[47], consequently, the substitution of the second residue would principally modulate the NTA, influencing the stability and/or biological function of the target protein. To further test the critical Nt-residues and the substrate specificity of NatB-mediated NTA of YUC8 in plants, we substituted the second E by A, a small aliphatic residue, generating a mutant form of YUC8(E2A). In vitro assay showed that the E2A substitution did not significantly alter the enzyme activity of YUC8 (Supplementary Figs. 11; 17), but greatly reduced the protein abundance in its over-expression plants (Fig. 4d, e; Supplementary Fig. 16) when compared with wild-type YUC8$^{ME}$-mGFP), causing a failure either to produce high auxin phenotypes, or to rescue the *yuc8* root phenotypes (Fig. 4a–c). These results again reveal that the intracellular stability of YUC8 for its steady-state level can largely depend on

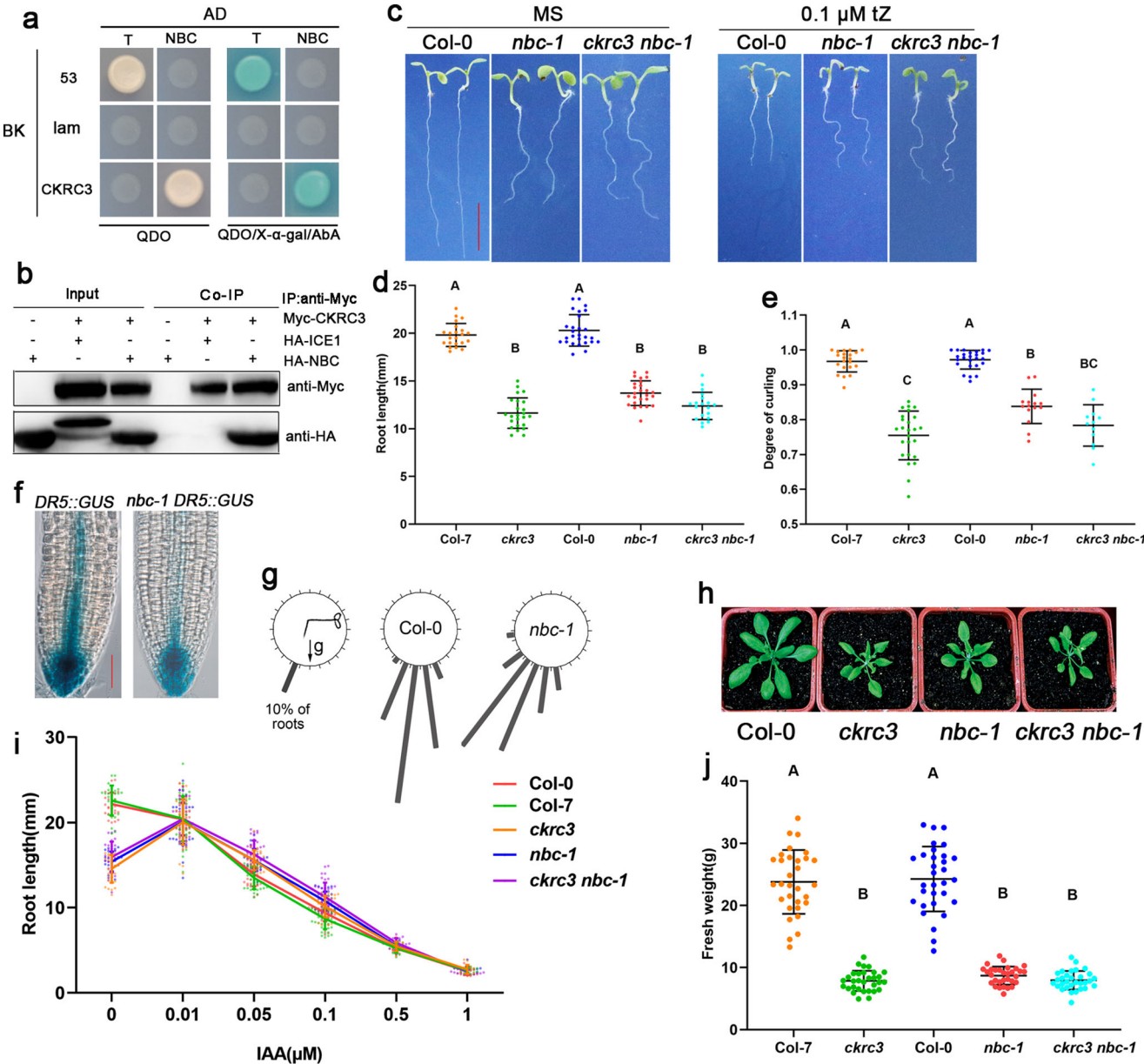

**Fig. 2 Evidences of interaction between CKRC3/TCU2 and NBC to form a NatB for maintaining endogenous auxin levels. a** Y2H test in vitro, with *p*GBKT7-53 and *p*GADT7-T as positive, *p*GBKT7-lam and *p*GADT7-T as negative controls, respectively. **b** Interaction between CKRC3 and NBC in vivo revealed by Co-IP test after transiently co-expressing *35S::*Myc-CKRC3 and *35S::*HA-NBC in *N.benthamiana* leaves. The nuclear protein ICE1 was used as negative control. **c–e** Plant phenotypes at 7 DAG; bar = 5 mm, $n \geq 12$. Data are presented as mean ± SD, different capital letters indicate significant differences at $P < 0.01$ according to ANOVA followed by Tukey's multiple comparison tests. **f** GUS staining of *DR5::GUS* marker lines, bar = 100 μm. **g** Quantification of root gravitropic responses ($n \geq 39$). **h, j** Rosettes and fresh weight of 30 DAG plant ($n = 31$). Data are presented as mean ± SD, different capital letters indicate significant differences at $P < 0.01$ according to ANOVA followed by Tukey's multiple comparison tests. **i** Primary root length at 7 DAG on MS medium with increasing concentrations of IAA ($n \geq 15$). Data are presented as mean ± SD.

the NatB activity. During the revision of this paper, Li et al (2020)[48] reported similar NatB-mediated stabilization of the stress/immune-related SIGMA FACTOR-BINDING PROTEIN1 (SIB1) in *Arabidopsis*.

## Conclusions

In summary, our present study reveals that in *Arabidopsis* NatB complex of CKRC3/TCU2-NBC can mediate the NTA of YUC8 for its intracellular stability to maintain auxin homeostasis (Fig. 4f) for normal growth. In *Arabidopsis*, 8 of the 11 YUCs possess NatB substrate signatures (Supplementary Fig. 7), catalyzing the rate-limiting step of the main auxin biosynthesis

pathway (IPyA pathway) in various tissue/organs[7,49]. As NatB is widely expressed in tissues throughout plant development (Supplementary Fig. 5a, c, d). NatB-mediated NTA of these YUCs must play significant roles in auxin homeostasis at the whole plant level for normal growth and development.

## Methods

**Plant material and growth conditions**. Information about the *Arabidopsis thaliana* mutants used in this study listed in Supplementary Tab. 2. The mutants were confirmed by PCR and sequencing. The primers used for PCR are given in Supplementary Tab. 3. Double mutant lines were obtained by crossing. For construction of transgenic vector, the full-length CDS of *YUC8* was amplified by RT-PCR from Col-0 and cloned into *pCAMBIA1302-mGFP* vector for generating

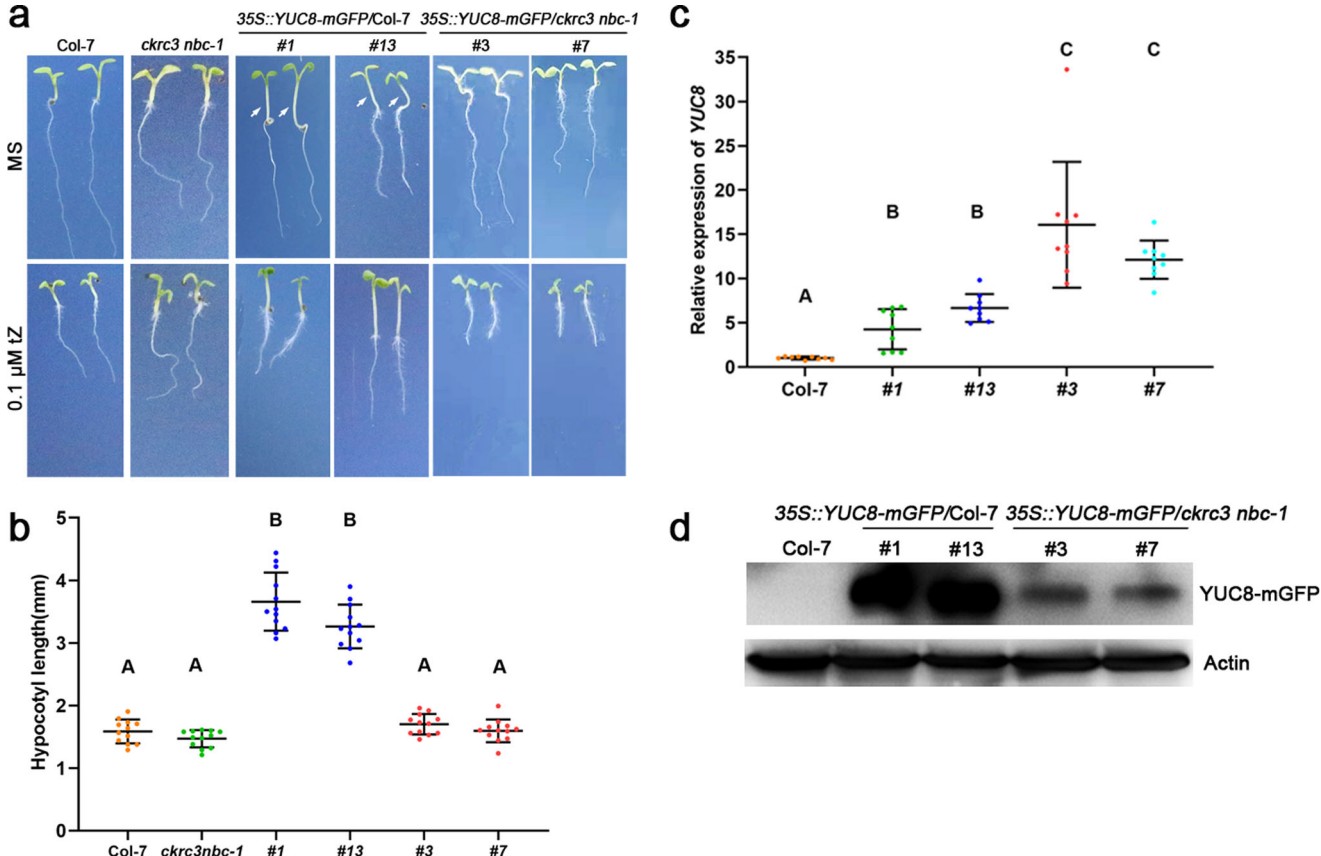

**Fig. 3 NatB participates in auxin homeostasis by modulating the stability of YUC8. a** Phenotypes of mutants and over-expression lines at 7 DAG on MS media with or without 0.1 μM tZ, bar = 5 mm. **b** Hypocotyl length at 7 DAG; $n = 12$. Data are presented as mean ± SD, different capital letters indicate significant differences at $P < 0.01$ according to ANOVA followed by Tukey's multiple comparison tests. **c** YUC8 expression levels determined by qRT-PCR. Data are presented as mean ± SD of three independent experiments, different capital letters indicate significant differences at $P < 0.01$ according to ANOVA followed by Tukey's multiple comparison tests. **d** YUC8-mGFP fusion protein levels determined by WB using anti-GFP antibody, ACTIN as internal reference.

$p35S::YUC8-mGFP$, $p35S::YUC8-mGFP/ckrc3$ $nbc-1$, $p35S::YUC8-mGFP/yuc8$ and $p35S::YUC8(E2A)-mGFP/yuc8$, $p35S::YUC8(E2A)-mGFP$/Col-7 transgenic lines. *Arabidopsis* transformation was performed by the *Agrobacterium tumefaciens*-mediated floral-dip method[50].

Seeds were sterilized with 0.1% HgCl₂, thoroughly washed with dH₂O three times, and placed on Murashige Skoog (MS) medium containing 1% agar and 1% sucrose. Before being transferred to culture under long-day conditions (16 h light/ 8 h dark) at 22 °C, the plates were kept at 4 °C for 2 days in darkness. After 7 days of growth, the plants were analyzed phenotypically or transferred to soil and cultivated in a greenhouse at the same growth conditions as above.

**Phenotype characterization**. For root growth inhibition assays and biochemical complementation, seeds were germinated and grown vertically on MS medium with different hormones at a selected range of concentrations at 22 °C with a 16/8 hour light/dark cycle for 7 days, then their root elongation was measured. All presented data were the mean values of three separate experiments using at least 60 seedlings for each replica.

The Degree of root Curling was calculated by dividing the distance between the two ends of a root ($L_0$) by its actual length ($L$).

For root gravitropic response assays, the germinated seedlings were first grown vertically on MS plates for 6 days at 22 °C with a 16/8 hour light/dark cycle, then transferred to fresh media. After 3 hours, the plates were rotated 90 degrees and grown horizontally for 24 hours. The degree of gravitropic response was measured for each root. Approximately 100 seedlings were measured for each genotype and treatment.

**Histochemical GUS assay**. For GUS staining, 7-day-old seedlings of Col/ Dr5::GUS, ckrc3/Dr5::GUS, nbc-1/Dr5::GUS grown on MS medium were harvested and incubated in 1 mM X-gluc (5-bromo-4-chloro-3-indolyl-β-D-glucuronide) at 37 °C for 40 minutes. The results were observed under light microscope.

**Constructs for subcellular localization**. The full-length coding sequences of *CKRC3/TCU2* and *NBC* were amplified by PCR using their respective specific pair of primers, and cloned via recombination technology into the *pCAMBIA2300* vector with the GFP coding sequence to construct *pCAMBIA2300-eGFP* and *pCAMBIA2300-GFP* vectors. GFP fusion proteins were expressed in tobacco cells 2 days after transient transformation. The GFP fluorescence was analyzed with a laser-scanning confocal microscope (Leica, https://www.leica.com/).

**RNA extraction and quantitative real-time PCR**. RNA was isolated using Trizol (Invitrogen, http://www.invitro-gen.com/) and reverse-transcribed using a reverse transcription kit (Takara, http://www.takara-bio.com/). Quantitative RT-PCR was performed in CFX (Bio-Rad, https://www.bio-rad.com/) real-time PCR equipment by using the SYBR green chemistry (Takara). *ACTIN8* was used as an internal control. Quantitative PCR analysis was performed with three different replicates for each biological sample. Three biological replicates were performed in each experiment.

**Y2H assays**. Y2H were performed according to the instructions provided with the Matchmaker LexA two-hybrid system (Clontech, http://www.takara-bio.com/). *pGBKT7-CKRC3/TCU2* constructs were obtained by inserting full-length coding sequences of *CKRC3/TCU2* into the EcoRI and SalI sites of the *pGBKT7* plasmid, and *pGADT7-NBC* constructs by inserting that of *NBC*. Corresponding pairs of plasmids were transformed into yeast strain. Yeast transformants were then plated on minimal SD/-Leu/-Trp agar plates and incubated for 4 days at 30 °C. The well-grown colonies were plated onto minimal SD/-Leu/-Trp/-His/-Ade agar plates with or without X-α-Gal and AbA for interaction tests for staining.

**Immunoprecipitation (IP)**. Full-length coding sequences of *CKRC3/TCU2-Myc* and *NBC-HA* were introduced into *pCAMBIA1302-mGFP* vector. IP was performed with transient expression of *CKRC3/TCU2-Myc* and *NBC-HA* constructs in *N. benthamiana* leaves, the protein ICE1 located in nucleus as a negative control. Leaves (0.3 g) were collected 24 h after infiltration of $4 \times 10^8$ cfu/mL, ground to a powder under liquid nitrogen, and resuspended in 2.0 mL of IP buffer containing 50 mM Tris, pH 7.5, 150 mM NaCl, 10% glycerol, 0.1% Nonidet P-40, 5 mM DTT, and 1.5×Complete Protease Inhibitor (Roche, https://www.roche.com.cn/). The

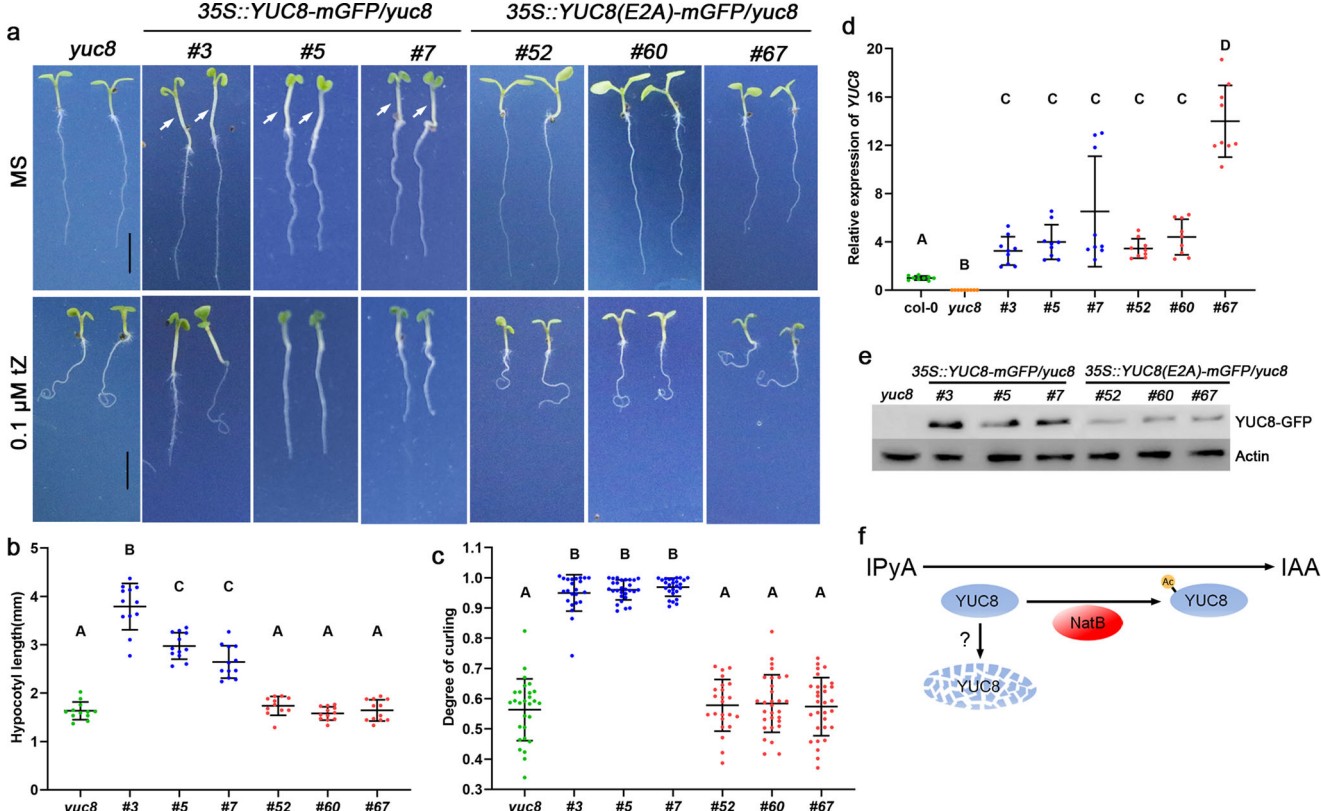

**Fig. 4 NatB-mediated NTA of YUC8 is determined by N-terminal Aa and is essential for its intracellular stability to function in auxin biosynthesis.**
**a** Phenotypes of *yuc8* mutant and transgenic *yuc8* plants harboring YUC8/YUC8(E2A)-mGFP at 7 DAG on MS media with or without 0.1 μM tZ, bar = 5 mm. **b**, **c** Quantification of Hypocotyl length (MS, $n \geq 11$) and degree of root curling (0.1 μM tZ, $n \geq 23$). Data are presented as mean ± SD, different capital letters indicate significant differences at $P < 0.01$ according to ANOVA followed by Tukey's multiple comparison tests. **d** *YUC8* expression levels determined by qRT-PCR. Data are presented as mean ± SD of three independent experiments, different capital letters indicate significant differences at $P < 0.01$ according to ANOVA followed by Tukey's multiple comparison tests. **e** YUC8-mGFP fusion protein levels determined by Western analysis using anti-GFP antibody, ACTIN as internal reference. **f** A proposed model elucidating the role of NatB in controlling the auxin biosynthesis.

crude lysates were then spun at 20,000×*g* for 10 min at 4 °C, and the supernatant was filtered through a 0.45-μm syringe filter. Filtered supernatant (0.75 mL) was diluted with 0.25 mL of IP buffer and used for each IP. Next, antibody (5 μL of either anti-Myc monoclonal antibody (1:5000; Novogene, https://www.novogene.com/, Catalog no. NHT0044) or anti-HA polyclonal antibody (1:5000; Proteintech, https://www.ptgcn.com/, Catalog no. 51064-2-AP) was used to capture the epitope-tagged proteins. The immunocomplexes were collected by adding 30 μL of protein G–Sepharose 4 fast flow (Amersham Pharmacia Biotech, https://www.hospitalnetwork.com/doc/amersham-pharmacia-biotech-inc-0001, Piscataway, NJ) beads and incubating end-over-end for 4 h at 4 °C. After incubation, the immunocomplexes were washed four times with 1 mL of IP buffer and the pellet was resuspended in SDS–PAGE loading buffer.

**Protein expression in prokaryotes and purification.** Full-length coding sequence of *NBC* was introduced into *pEASY-Blunt E2-His-MBP* vector. The *pEASY-Blunt E2-His-MBP-NBC* plasmid was transformed into *E. coli* BL21 Star (DE3) cells (Invitrogen). Cell culture was grown in LB (lysogeny broth) medium to an $OD_{600nm}$ of 0.6 at 37 °C and subsequently transferred to 18 °C. After 30 min of incubation, protein expression was induced by isopropyl-β-D-thiogalactoside (0.5 mM). After 12 h of incubation, the cultures were harvested by centrifugation (4000 rpm) at 4 °C. *E. coli* pellets containing recombinant proteins were lysed by sonication in lysis buffer (1 mM DTT, 50 mM Tris-HCl (pH 7.4), 300 mM NaCl, 1 tablet EDTA-free protease Inhibitor cocktail per 50 ml (Roche). The cell extracts were applied on a Ni-TED 1 ml Sefinose (TM) Column (Sangon Biotech, https://www.sangon.com/). The purity of the fractions corresponding to purified monomeric recombinant proteins was analyzed on Coomassie-stained SDS-polyacrylamide gel electrophoresis (SDS–PAGE) gels and the protein concentrations were determined by $OD_{280nm}$ measurements.

**In vitro NAT assay.** Purified His-MBP-NBC (0.5 mM) was mixed with selected oligopeptide substrates (200 mM) and 300 mM of acetyl-CoA in a total volume of 60 μl acetylation buffer (50 mM Tris-HCl, pH 7.5, 10% glycerinum, 10 mM DTT, 1 mM EDTA). The samples were incubated at 37 °C for 60 minute. The enzyme

activities were quenched by adding 5 ml of 10% TFA. The acetylation reactions were quantified using RP-HPLC and LC-MS/MS. Synthetic peptides were custom-made to a purity of 90%. All peptides contain 10 unique amino acids at their N-terminus, as these are the major determinants influencing NTA. The next 17 amino acids are essentially identical to the ACTH peptide sequence (RWGRPVGRRRRPVRVYP) except that the lysines were replaced by arginines to minimize any potential interference by N-terminal acetylation. Peptide sequences: SNC1, MDTSKDDDMERW GRPVGRRRRPVRVYP; YUC8, MENMFRLMDQRWGRPVGRRRRPVRVYP; YUC8 (MAE), MAENMFRL MDQRWGRPVGRRRRPVRVYP.

**Plant protein extraction.** Plant tissue was ground in liquid nitrogen to a fine powder. The protein extraction buffer (50 mm Tris-HCl [pH 7.5], 150 mm NaCl, 5 mm DTT, 10% [v/v] glycerol, 1% [v/v] Nonidet P-40, and 1×Complete protease inhibitor cocktail) was added at a 1:3 ratio (tissue:buffer) to the powder and mixed well. The supernatant was harvested after two centrifugations at 13,000 rpm for 20 min in a 4 °C table-top centrifuge. Then the protein content of all samples was quantified. The resulting proteins were resuspended in 2× SDS sample buffer and denatured for 10 min at 98 °C.

**In vivo NAT assay.** IP Coupled with MS Analysis. Total proteins were isolated from transgenic plants overexpressing YUC8N7::eYFP fusion protein using the protein extraction buffer described above. The protein extract was incubated with 25–35 μL of anti-GFPmAb-magnetic beads (Chromotek, https://www.chromotek.com/) for 4 h at 4 °C with constant rotation. The beads were then washed three times with the IP buffer. After the last wash, 25 μL of the beads was eluted in 50 μL of 2× SDS protein sample buffer by incubating at 98 °C for 10 min. The eluates were separated by 12% SDS–PAGE, and detected by immunoblot analyses using a mouse anti-GFP monoclonal antibody (1:5,000; Abmart, http://www.ab-mart.com.cn/, Catalog no. M20004). Cutting the target bands from IP into cubes (ca. 1 ×1 mm). The proteins were reduced, alkylated, and destained in gel, saturated, and digested by trypsin for LC-MS/MS (nanoLC-MS/MS, Orbitrap Fusion™ Lumos™ Tribrid™) analysis. The mass spectra were submitted to the Proteome Discoverer (Thermo Scientific, https://www.thermofisher.cn/cn/zh/home.html) for peptide identification.

**SDS–PAGE and western blot**. Protein samples were separated by SDS–PAGE and then transferred to a polyvinylidene fluoride film. After blocking with 5% bovine serum albumin, the film was incubated with primary antibody overnight at 4 °C, then washed three times with PBST for 10 minute and incubated with secondary antibody for 1 hour at room temperature. After washing three times with PBST for 10 minute, the film was illuminated using a luminous imaging system. The YUC8-mGFP fusion protein was immunochemically detected using a mouse anti-GFP monoclonal antibody (1:2000; Abmart, Catalog no. M20004M). ACTIN protein was detected using mouse anti-Plant Actin monoclonal antibody (1:5000; Abbkine, https://www.abbkine.com/, Catalog no. ABL1050).

**Transient expression in protoplasts**. The transient expression was performed as described[51]. The *YUC8-eGFP/YUC8 (E2A)-eGFP* CDS was amplified by PCR and inserted into *pCAMBIA2300* vector. Primers used are listed in Supplementary Tab. 3. For transient expression assay, *Arabidopsis* mesophyll protoplasts from the Col-7 and *ckrc3 nbc-1* mutant were transfected with 200 μg plasmid and incubated 18 h. The GFP fluorescence was observed under a laser scanning confocal microscope Zeiss 880 (Zeiss, https://www.zeiss.com/corporate/int/home.html).

**YUC8 enzyme activity assay**. Protein expression and purification for YUC8/YUC8(E2A) was the same as that for NBC, and the His-MBP-YUC8 fusion protein was detected using an anti-His polyclonal antibody (1:5000; Solarbio, https://www.solarbio.com/, Catalog no. k007439p). To test the enzyme activity, about 2 μg YUC8 protein (estimated in SDS–PAGE gel by comparison with the protein marker), NADPH (50 mM) 20 μL, FAD (2 mM) 2 μL, IPyA (50 mM) 0.4 μL and add nuclease-free water to a final volume of 100 μL, without substrate IPyA or YUC8/YUC8(E2A) as control (CK1 or CK2), the mixture was incubated at 30 °C for 2 hours with vigorous shaking. IAA was quantitatively analyzed by HPLC (high-performance liquid chromatograph) in Suzhou Keming Biotechnology Co. LTD (http://sunnyblw.bioon.com.cn/), the IAA values were quantified using a standard curve ($y = 9.959x - 2.537$; $R^2 = 0.9999$).

**Statistics and reproducibility**. All results are expressed as the means ± standard deviation. The numbers of samples and replicates of experiments were shown as mentioned in the figure legends. Comparisons between groups were determined using ANOVA followed by Tukey's multiple comparison test. All data were analyzed by using GraphPad Prism 8 software (https://www.graphpad.com/).

**Accession numbers**. Accession numbers of the genes mentioned in this study are *CKRC3/TCU2* (AT5G58450), *NBC* (AT1G03150), *CKRC1/TAA1* (AT1G70560), *SUR2* (AT4G31500), *YUC1* (AT4G32540), *YUC2* (AT4G13260), *YUC5* (AT5G43890), *YUC6* (AT5G25620), *CKRC2/YUC8* (AT4G28720), *YUC9* (AT1G04180), *YUC10* (AT1G48910), *YUC11* (AT1G21430), *CYP79B2* (AT4G39950), and *CYP79B3* (AT2G22330).

**Reporting summary**. Further information on research design is available in the Nature Portfolio Reporting Summary linked to this article.

## Data availability
All data generated during this study are included in the supplementary information files. Original statistical data for all Figures and Supplementary Figures are shown in Supplementary Data.

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

## Acknowledgements

We acknowledge the excellent technical support from the Life Science Research Laboratory Center at Lanzhou University. This work was supported by grants from the National Natural Science Foundation of China (grant numbers 31671458 and 31970713).

## Author contributions

G.Q.G. conceived and directed the research; H.Q.L. performed the major part of the research; Z.X.P. & Y.J.Z. did in vivo NatB activity analysis; Z.X.P. and L.W. repeated most critical experiments; D.W.D. isolated the *ckrc3* mutant and cloned the gene; Y.M.G. did Y2H and Co-IP test; J.L.W., Z.Z., P.T., and Q.H.F. participated in phenotype analysis. A.J.K. and X.F.L. participated in the discussion; H.Q.L., Z.X.P., L.W., and G.Q.G. analyzed the data and coordinated the figures. L.W. assisted G.Q.G. to organize and coordinate the project. H.Q.L. wrote the draft of the manuscript and the final revision was accomplished by L.W. and G.Q.G.; all authors discussed the data and the article.

## Competing interests

The authors declare no competing interests.
