## [Peer Review File · Communications Biology]

Reviewers' comments:

Reviewer #1 (Remarks to the Author):

In this manuscript, the authors discovered that disruption of CKRC3 leads to partial auxin deficiency. Furthermore, they showed N-terminal acetylation of YUC affects YUC stability and auxin homeostasis. The work is very interesting and demonstrates the importance of posttranslational modification in regulating auxin biosynthesis.

The manuscript was written in a very short format, rendering it very difficult to read. Why not submit it as a full-length article? Many results were not discussed. For example, in figure 1k, the authors stated that *ckrc3* was rescued by *sur2*. To me, *sur2* was rescued by *ckrc3*. This is highly significant and also confusing. The high auxin phenotype of *sur2* is caused by defects in glucosinolate biosynthesis. Such a perfect rescue is interesting because SUR2 and YUC are in totally different pathway. The results should be discussed.

In figure 3a, I do not understand why 35S::YUC8-mGFP/*ckrc3* would be more sensitive to tZ than both WT and the overexpression line in Col in root elongation?

Fig. 4a, the E2A plants seem more sensitive to tZ in terms of plant size.

Overall, the results are interesting, but the manuscript is difficult to read because of the extremely abbreviated format.

Reviewer #2 (Remarks to the Author):

The paper by Liu et al. reports the characterization of a point mutation in the At5g58450 gene, corresponding to the auxiliary subunit (Naa25) of Arabidopsis NatB. This mutation causes a premature termination at the 731th aa position of the encoded protein. The authors, in the frame of a large-scale screening to identify novel auxin-deficient mutants, have previously isolated this mutant. The authors provide data, suggesting a link between protein NTA and Auxin homeostasis in plants.

While I am excited by this kind of data and I'm sure they will be useful for others that are interested in acylation in plants, I feel that the reported results are both incomplete and not convincing. To a considerable degree, the findings presented are descriptive and many conclusions are misleading. Several key questions need to be still addressed to make this article suitable for publication.

Major concern:

- 1) Phenotypic changes associated with point mutation in the Naa25 subunit might be directly or indirectly linked to N-terminal acetylation of NatB. It is totally missed the N-terminomics data related to all different mutant lines are reported.
- 2) No evidence is shown this Naa25 mutation is responsible via reduced NatB activity (not shown either, see point 1) of half-life of YUC8. Low accumulation of a protein can be due to many reasons (i.e., low transcripts, reduced protein synthesis ecc...).
- 3) What happens if YUC8(E2A) is overexpressed in *ckrc3* and *ncbc-1* double and single mutants? I missed the logic of experiment in Fig. 4 and the claimed conclusions.

Minor concerns:

- 1) Many abbreviations are not reported. The understanding of figures and particularly their legends is very difficult.
- 2) Legend description are not informative and does not allow to understand the figures.
- 3) Several Typos should be corrected (i.e., AVONA with ANOVA).

Reviewer #3 (Remarks to the Author):

The manuscript by Liu et al. provides new insights into the critical role of NatB in auxin homeostasis and demonstrated that NatB functions through post-translational regulation of YUC8, a key enzyme in auxin biosynthesis. Their work is interesting in that they provided a new level of regulation in auxin biosynthesis. Here are a few concerns I have.

1. *ckrc3/naa25* exhibited pleiotropic defects in plant development, which may result from altered regulation on other targets of NatB. Thus, instead of saying “*ckrc3* mutant exhibited a number of typical auxin-deficient phenotypes...”, the authors should clearly demonstrate which of the phenotypes are resulted from auxin deficiency either through application of exogenous auxin or complementation using *sur2*. And later, using these phenotypes to examine the effect of YUC8 over-expression lines. It would also be nice to directly measure the auxin level in the mutant to rule out other possibilities such as a defect in auxin transport.
2. The author showed that root length and the degree of curling (on 1/2MS I guess) are two phenotypes that can be rescued by IAA or *sur2*. However, in Fig. 3C, degree of curing in response to tZ was shown. Why is that? I think the author should consistently use root curling on 1/2MS as a phenotypic measurement. Although over-expressing YUC8-mGFP in *ckrc3nbc1* mutant did not produce long hypocotyl phenotype, it does seem to rescue its root curling phenotype in response to tZ, which is confusing. The author did not mention Fig 3C in the main text and a Col-0 control is missing here. Also, both over-expressing YUC8-mGFP in wt and mutation *ckrc3nbc1* produced short root phenotype, it is thus hard to conclude whether over-expressing YUC8-mGFP in *ckrc3nbc1* mutant rescued its phenotype or not.
3. Although the author demonstrated that YUC8 is a target of NatB in vitro, it is still critical to prove that YUC8 can be modified in vivo as many enzymes display non-specific enzymatic activity in vitro.
4. Although YUC8(E2A) mutation clearly affected YUC8 accumulation, significant amount of YUC8 protein can still be detected. It is thus surprising that 35S::YUC8(E2A) did not rescue *yuc8* mutation at all. Does the mutation also affect the enzymatic activity of YUC8? A 35S::YUC8-mGFP/*yuc8* with low expression level (comparable to the YUC8(E2A) over-expressing lines) could be used to rule out this possibility
5. In Fig. 2b, an unrelated protein should be used as a negative control.
6. Fig. 1g: DR::GUS should be DR5::GUS. In material and methods, the author used Dr5::GUS. The format should be consistent.

Dear reviewers ,

Thanks for the comments from Reviewers. Our reply is as follows:

Reviewer #1

1. The manuscript was written in a very short format, rendering it very difficult to read. Why not submit it as a full-length article? Many results were not discussed. For example, in figure 1k, the authors stated that *ckrc3* was rescued by *sur2*. To me, *sur2* was rescued by *ckrc3*. This is highly significant and also confusing. The high auxin phenotype of *sur2* is caused by defects in glucosinolate biosynthesis. Such a perfect rescue is interesting because *SUR2* and *YUC* are in totally different pathway. The results should be discussed.

Reply: The explanation about the role of *sur2* mutation in rescuing *ckrc3* was added in the text.

2. In figure 3a, I do not understand why 35S::*YUC8*-mGFP/*ckrc3* would be more sensitive to tZ than both WT and the overexpression line in Col in root elongation?

Reply: It was indeed the case, in terms of root length and root hairs, as shown in the original picture below. It should be noted that *YUC8* protein can still be detected in *ckrc3 nbc-1* background, although relatively weak (Fig3e). In the presence of exogenous tZ, root phenotype may be more sensitive to auxin perturbation, and at the moment we do not know why. However, *YUC8* over-expression-mediated auxin phenotypes were obvious in hypocotyl and cotyledon, which is consistent with the role of NatB-mediated NTA on the protein stability.

3. Fig. 4a, the E2A plants seem more sensitive to tZ in terms of plant size.

Reply: This was true. The average plant size of the *35S::YUC8(E2A)mGFP/yuc8* is slightly larger than that of *yuc8* plants on MS, but comparable between the two genotypes on MS+tZ.

Reviewer #2

Major concern:

1. Phenotypic changes associated with point mutation in the Naa25 subunit might be directly or indirectly linked to N-terminal acetylation of NatB. It is totally missed the N-terminomics data related to all different mutant lines are reported.

Reply: To us it seems unlikely that comparing N-terminomics data between different mutant lines can be very helpful to determine the association between the mutant phenotypes and the NatB activity, since many actual protein (such as YUC8) NTA are not covered by this technique, as reported in a recent N-terminomic study on *natb* mutants (Huber M, Bienvenu WV, Linster E, Stephan I, Armbruster L, Sticht C, Layer D, Lapouge K, Meinnel T, Sinning I, Giglione C, Hell R, Wirtz M. NatB-Mediated N-Terminal Acetylation Affects Growth and Biotic Stress Responses. *Plant Physiol.* 2020 Feb; 182(2): 792-806. doi: 10.1104/pp.19.00792.). Moreover, N-terminomic analysis is too expensive for us to accomplish such a costly but un-urgent works; nevertheless, our *in vitro/vivo* assay (Extended data Fig.9 and 12) on NatB activity clearly show that *natb* mutations affect YUC8 protein NTA to produce auxin-related phenotypes.

2. No evidence is shown this Naa25 mutation is responsible via reduced NatB activity (not shown either, see point 1) of half-life of YUC8. Low accumulation of a protein can be due to many reasons (i.e., low transcripts, reduced protein synthesis etc...)

Reply: As show in Fig3d-e, we have detected the transcription level and protein level of YUC8 in the transgenic line. The results showed that, compared with that in the WT background, the transcription level of YUC8 was even higher in the double mutant *ckrc3 nbc-1* background (fig3d), but the protein level of YUC8 was lower (fig3e). This result confirmed that low accumulation of YUC8 cannot be due to low transcripts or translation etc.

3. What happens if YUC8(E2A) is overexpressed in *ckrc*” *nbc-1* double and single mutants? I missed the logic of experiment in Fig. 4 and the claimed conclusions.

Reply: The NatB-mediated NTA of YUC8 depends on the second N-aa of E. We checked that in the *yuc8* mutant rather than *natb* mutants (defective in NatB activity).

Minor concerns:

- 1) Many abbreviations are not reported. The understanding of figures and particularly their legends is very difficult.
- 2) Legend description are not informative and does not allow to understand the figures.
- 3) Several Typos should be corrected (i.e., AVONA with ANOVA).

Reply: The figures and legend description have been revised, and other writing errors have been corrected.

Reviewer #3

1. *ckrc3/naa25* exhibited pleiotropic defects in plant development, which may result from altered regulation on other targets of NatB. Thus, instead of saying “*ckrc3* mutant exhibited a number of typical auxin-deficient phenotypes...”, the authors should clearly demonstrate which of the phenotypes are resulted from auxin deficiency either through application of exogenous auxin or complementation using *sur2*. And later, using these phenotypes to examine the effect of YUC8 over-expression lines. It would also be nice to directly measure the auxin level in the mutant to rule out other possibilities such as a defect in auxin transport.

Reply: Reduced length of primary roots, root curling, defective gravitropic response are typical auxin-deficient phenotypes, and can be rescued either by exogenous application or endogenous production (*sur2* mutation) of auxin (manuscript Fig. 2i-I, Extended Data Fig.2). Moreover, visual observations on the typical YUCs-mediated high auxin phenotypes

with long hypocotyls and epinastic cotyledons were also used to evaluate the phenotypic effects of YUC8 levels in plants.

The reduced auxin level in *ckrc3* has been measured in our previous works (Wu, L. *et al.* 2015. Forward genetic screen for auxin-deficient mutants by cytokinin. *Scientific reports* 5, 11923).

2. The author showed that root length and the degree of curling (on 1/2MS I guess) are two phenotypes that can be rescued by IAA or *sur2*. However, in Fig. 3C, degree of curing in response to tZ was shown. Why is that? I think the author should consistently use root curling on 1/2MS as a phenotypic measurement. Although over-expressing YUC8-mGFP in *ckrc3nbc1* mutant did not produce long hypocotyl phenotype, it does seem to rescue its root curling phenotype in response to tZ, which is confusing. The author did not mention Fig 3C in the main text and a Col-0 control is missing here. Also, both over-expressing YUC8-mGFP in WT and mutation *ckrc3 nbc1* produced short root phenotype, it is thus hard to conclude whether over-expressing YUC8-mGFP in *ckrc3nbc1* mutant rescued its phenotype or not.

Reply: See our replies to the above question and the 2nd question of reviewer #1.

3. Although the author demonstrated that YUC8 is a target of NatB *in vitro*, it is still critical to prove that YUC8 can be modified *in vivo* as many enzymes display non-specific enzymatic activity *in vitro*.

Reply: During revision we did this measurement and the *in vivo* NatB activity are presented in Extended data Fig.12.

4. Although YUC8(E2A) mutation clearly affected YUC8 accumulation, significant amount of YUC8 protein can still be detected. It is thus surprising that 35S::YUC8(E2A) did not rescue *yuc8* mutation at all. Does the mutation also affect the enzymatic activity of YUC8? A 35S::YUC8-mGFP/*yuc8* with low

expression level (comparable to the YUC8(E2A) over-expressing lines) could be used to rule out this possibility.

Reply: During revision we repeated the experiment and the results are shown in Fig. 4e. The abundance of YUC8 fusion protein in *35S::YUC8(E2A)-mGFP/yuc8* is very low compared with *35S::YUC8-mGFP/yuc8*. More significant amount of YUC8 appeared in the previous WB picture may be caused by too much long exposure time.

We measured enzyme activity of YUC8 and YUC8(E2A) in *vitro*, and found that E2A mutation did not significantly affected enzyme activity (revised manuscript Extended data Fig.11).

5. In Fig. 2b, an unrelated protein should be used as a negative control.

Reply: During revision we have used the protein ICE1 as a negative control (revised manuscript Fig 2b).

6. Fig. 1g: DR::GUS should be DR5::GUS. In material and methods, the author used Dr5::GUS. The format should be consistent.

Reply: corrected.

Sincerely yours,

Guang-Qin Guo (corresponding: gqguo@lzu.edu.cn)

MOE Key Laboratory of Cell Activities and Stress Adaptations, Lanzhou University, Lanzhou 730000, China

Referee expertise:

Referee #1: The authors have adequately addressed my concerns.

Referee #2: The authors have carefully addressed most of my questions. I still have a little concern over the YUC8-over expression lines. Although the effect of YUC8-ox on hypocotyl growth is clearly CKRC-dependent, its effect in root is not straightforward. As over-expression of YUC8 in wt generates short root, one cannot conclude if over-expressing YUC8 rescues the short root of *ckrc3nbc1* or not. Over-expression of YUC8 in wt and *ckrc3nbc1* also seems to generate similar root curling phenotype under both control and tZ treatment, it is thus hard to explain the result by arguing the increased auxin sensitivity in tZ. These defects may indeed be auxin-dependent, but whether it is YUC8-dependent or not would be difficult to tell.

I would suggest using hypocotyl phenotype of *ckrc3* and *yuc8*, for example high temperature-induced elongation growth, as a YUC8-dependent phenotypic marker, and examine this phenotype for both YUC8-ox and YUC8-E2A lines.

Minor concerns:

Fig. 1a and k, the same figure for *ckrc3* in tZ treatment was used.

In Fig. 1l, it was not clear the root curling was measured under what condition.

Fig.3b and 3c were not mentioned in the main text, and there is no explanation about the results.

Dear reviewers ,

Thanks for the comments from Reviewers. Our reply is as follows:

Referee #1: The authors have adequately addressed my concerns.

Referee #2: The authors have carefully addressed most of my questions. I still have a little concern over the YUC8-over expression lines. Although the effect of YUC8-ox on hypocotyl growth is clearly CKRC-dependent, its effect in root is not straightforward. As over-expression of YUC8 in wt generates short root, one cannot conclude if over-expressing YUC8 rescues the short root of *ckrc3nbc1* or not. Over-expression of YUC8 in wt and *ckrc3nbc1* also seems to generate similar root curling phenotype under both control and tZ treatment, it is thus hard to explain the result by arguing the increased auxin sensitivity in tZ. These defects may indeed be auxin-dependent, but whether it is YUC8-dependent or not would be difficult to tell.

I would suggest using hypocotyl phenotype of *ckrc3* and *yuc8*, for example high temperature-induced elongation growth, as a YUC8-dependent phenotypic marker, and examine this phenotype for both YUC8-ox and YUC8-E2A lines.

In our study, we found the hypocotyl length is a good phenotype for testing the YUC8 function by its overexpression, and in that sense our presented results in Figs 3 and 4 clearly showed that the overexpressed YUC8 level and activity were dependent on NatB mediated NTA. Meanwhile, just going to look temperature-induced hypocotyl elongation of *ckrc3* and *yuc8* or YUC8-ox and YUC8(E2A)-ox lines (YUC8/E2A-ox is constitutive rather temperature-dependent) are unlikely to get better results. During revision days, we checked that by comparing the temperature-induced hypocotyl elongation between WT and related mutants/transgenics. As shown in the present Figure below, temperature-induced hypocotyl elongation was observed both in WTs

and *yuc8* single mutant, fitting the fact that *YUC8* is specifically expressed in roots but not in hypocotyl, and that other hypocotyl expressed YUCs are functional in temperature-induced elongation. Such responses were significantly reduced in *ckrc3/nbc-1* mutants, most likely due to the defective NatB-mediated NTA of those YUCs (we detected in vitro NTA of all the other YUCs with NatB substrate signature, data not present here). Despite of the lack of apparent temperature responses in various *YUC8* overexpression lines, both the *ckrc3/nbc* mutation or *YUC8(E2A)* substitution significantly reduced the hypocotyl elongations in both temperatures, indicating that NatB is required for the overexpressed *YUC8* to function to stimulate hypocotyl elongation. In summary, our present supplement experiment reconfirmed our conclusion that NatB-mediated NTA can stabilize YUCs for their function in auxin biosynthesis for plant growth and development.

Figure. Hypocotyl length after 7 days of long day at different temperatures. “**” indicate significant differences at $P < 0.01$, according to student t-tests. Mean \pm SD.

Minor concerns:

Fig. 1a and k, the same figure for ckrc3 in tZ treatment was used.

Indeed, the results of Fig1a and 1b are simply repeats or duplication of those of the Figs 1k and 1j, so we deleted Fig1a and 1b in this revision.

In Fig. 1l, it was not clear the root curling was measured under what condition.

The root curling in Fig1l (1d in this revision) was measured on MS without tZ, which was described in the present version.

Fig.3b and 3c were not mentioned in the main text, and there is no explanation about the results.

The results in these two figures are not closely relevant to the main content of the article, so we deleted them.

Sincerely yours,

Guang-Qin Guo (corresponding: gqguo@lzu.edu.cn)

MOE Key Laboratory of Cell Activities and Stress Adaptations, Lanzhou University, Lanzhou 730000, China